# A Comparison of Aerodynamic Parameters in Two Subspecies of the American Barn Owl (*Tyto furcata*)

**DOI:** 10.3390/ani12192532

**Published:** 2022-09-22

**Authors:** Hermann Wagner, Paolo Michael Piedrahita

**Affiliations:** 1Institute of Biology II, RWTH Aachen University, Worringerweg 3, D-52074 Aachen, Germany; 2Facultad de Ciencias de la Vida, Escuela Superior Politécnica del Litoral, Guayaquil P.O. Box 09-01-5863, Ecuador

**Keywords:** wing loading, flight, aspect ratio, reversed sexual dimorphism, allometry

## Abstract

**Simple Summary:**

Morphology and function depend on the ecological niche in which an animal lives. Barn owls, occurring on all continents, occupy a nocturnal niche. These birds prey mainly on small rodents but include other small vertebrates and invertebrates in the diet. The size of the barn-owl species and subspecies varies considerably. The American continent harbors the species *Tyto furcata*. The body mass of the subspecies in North America (*T.f.pratincola*) is about a factor of two higher than that of the subspecies living on the Galapagos archipelago (*T.f.puncatissima*). We asked how this difference translates into aerodynamic parameters. The key question was whether there is so-called similarity scaling or not. In other words, whether important aerodynamic parameters scale according to body mass. This is called isometric scaling. Deviation from isometric scaling is called allometric scaling. If we use the subspecies from the continent as a reference, we find that the Galapagos barn owl has relatively larger wings than expected from isometric scaling. This translates into a lower wing loading in *punctatissima* than in *pratincola*. A lower wing loading means higher maneuverability. We speculate that the higher maneuverability allows the Galapagos owl to catch smaller prey, especially insects.

**Abstract:**

Aerodynamic parameters, such as wing loading, are important indicators of flight maneuverability. We studied two subspecies of the American Barn owl (*Tyto furcata*), the North American subspecies, *T.f.pratincola*, and the Galapagos subspecies, *T.f.punctatissima*, with respect to aerodynamic parameters and compared our findings with those in other owl and bird species. The body mass of *T.f.pratincola* is about two times higher than that of *T.f.punctatissima*. Wing loading between the two subspecies scales allometrically. Wing loading in *T.f.pratincola* is about 50% higher than in *T.f.punctatissima*. The scaling of wing length is not statistically different from the prediction for isometric scaling. By contrast, the wing chord in *T.f.punctatissima* is larger than predicted by isometric scaling, as is the wing area. The scaling of wing loading observed here for *T.f.punctatissima* differs considerably from the scaling in other owl and bird species as available in the literature. We speculate that the allometric scaling helps *T.f.punctatissima* to catch smaller prey such, as insects that are found in many pellets of *T.f.punctatissima*, despite the fact that in both subspecies, small rodents make up most of the diet.

## 1. Introduction

Barn owls (*Tyto* sp.) are interesting when studying flight [1,2,3,4]. On the one hand, their aerodynamic adaptations are interesting [5,6,7]. On the other hand, the development of silent flight in these animals has been an inspiration for engineering applications [8,9,10]. We concentrate here on the aerodynamic adaptations. The comparative study by Rayner [11] demonstrated that wing loading in the European barn owl (*Tyto alba*) is in the lower range compared with other bird species. A similar conclusion may be derived for the North American barn owl from the data in [12]. Moreover, the body mass of barn owls is comparatively low [13]. By contrast, the wing area in these birds is relatively large, while the aspect ratio is low [12,13,14,15,16]. A low wing loading is advantageous because it increases maneuverability. A low body mass and a large wing area allow slow flight. Slow flight produces less noise. Moreover, a recently observed flow-turning effect in the boundary layer of the barn owl wing serves to attenuate crossflow instabilities and delay transition [17]. In other words, the data in [5,6,17] demonstrate that the owl wing has mechanisms to stabilize flow that improve aerodynamic behavior but also contribute to silent flight.

The genus Tyto has about 50 taxa [18,19,20]. Subspecies, such as the North American barn owl (*Tyto furcata pratincola*) and the Galapagos barn owl (*Tyto furcata punctatissima*), have very similar habitus and general behavior [18,21,22,23]. These two subspecies are referred to in the following text as *T.f.pratincola* and *T.f.punctatissima*, respectively. Despite the similar habitus, the body-mass difference in these two subspecies is large. This provides the opportunity to study, on the one hand, how body-mass difference translates into aerodynamic parameters in two closely related taxa and, on the other hand, whether *T.f.punctatissima*, living on remote islands, might show some specific adaptations.

These issues may be tackled by determining the important aerodynamic parameters and by checking whether they scale isometrically or allometrically. With respect to aerodynamic parameters, theoretical expectations may be derived from the relation of several parameters with body mass. If we accept the simple model that body mass scales with the third power of length and with the second power of area, we can test whether the relations are isometric or allometric [24].

Thus, the main aim of this study was to provide the first data ever on the aerodynamic parameters for *T.f.punctatissima*. A further aim was to compare the data of *T.f.punctatissima* with the data of *T.f.pratincola*. Finally, we wanted to find out how these parameters scale in these two subspecies. In the following, we show that despite the similar habitus and behavior, many aerodynamic parameters scale allometrically in the two subspecies *T.f.pratincola* and *T.f.punctatissima* and discuss why this may be so.

## 2. Materials and Methods

This study is based on data from two subspecies of the American barn owl (*Tyto furcata*). On the one hand, data from 10 adult specimens of the North American barn owl (*Tyto furcata pratincola*), four males and six females, were included (sample “*pr*”). These birds are descendants of specimens that originally came from California (USA), which had been held and bred in captivity for more than 30 years and stemmed from the colony at the Institute of Biology II at RWTH Aachen University, Aachen, Germany. This colony was kept under a permit of the Umweltamt Aachen and a permit for animal experiments of LANUV, Recklinghausen (Germany). The second source of data came from 7 living animals of the Galapagos barn owl (*Tyto furcata punctatissima*), three males and four females (sample “*pu*”). This sample includes the data from the wings of specimens that were studied in successive years. For the determination of the body mass, the data of four more birds (1 male and 3 females) was available. While we averaged the body mass of the birds that were caught and weighted several times (for details, see Appendix A), we only took information from the, in our view, most informative photo for the analysis of the wing parameters. The owls were studied under permits issued by the Galapagos National Park (PC-19-16; PC-22-17; PC-28-18; PC-73-19). The birds were caught, physically examined, ringed, and released. The data were collected between 2016 and 2019. No harm was inflicted on the living animals during the examinations, neither on Galapagos nor in Aachen.

We were interested in the following aerodynamic parameters of owl flight: 

Wing length l (unit meter (m)): We defined wing length as the distance from the base (either the border of the trunk or the inner margin of the 14^th^ or 15^th^ secondary, whichever was the most clearly visible) to the tip of the wing and measured it parallel to the leading edge of the wing. Although this is different from the standard ornithological morphometrics [25], we felt that this way of measuring wing length is justified in the barn owl. The reason is that the wing is swept back distally (see also [4,17] so that the line from the tip of the wing (the tip of the tenth primary) to the base is almost parallel to the leading edge (Figure 1a,b).

Wing length has to be calculated from the wing as extended during flight [25]. We used an image of a flying barn owl extracted from a video (Figure 1a) as well as a prepared wing (not shown) as references. The image of the flying barn owl demonstrated that the leading edge (the anterior edge of the wing that is directly hit by the airflow) was perpendicular to the body, and only the outermost part of the wing was swept back. In the process of reconstructing wing length, we adjusted a given wing on a photograph to the appearance as had been observed in the reference. This was possible by dividing the wing into several straight segments along its long axis. In other words, if a wing on a photograph was not stretched properly, we divided the long axis into up to four straight segments of different orientations from the base to the tip of the wing (Figure 1c). The length of each of the straight segments was measured. The lengths of the individual segments were added to arrive at the veridical wing length. For example, a length of 0.3 m was measured in the field for the wing shown in Figure 1c (the red line in Figure 1c shows how wing length was measured with a ruler in the field). When analyzing the photograph, it became clear by looking at the positions of the primary and secondary flight feathers that the wing was not appropriately stretched. On the one hand, the leading edge close to the body was not photographed. This was corrected by drawing a line from the base of the 14^th^ secondary (the innermost, 15^th^, the secondary was not visible) perpendicular to the proximal part of the leading edge and measuring the proximal part of the long axis from the intersection point to the point where the wing started to angle (proximal green line in Figure 1c). Three more segments were distinguished more distally (distal green lines in Figure 1c). Distally, the leading edge was swept back by about 30 degrees with respect to the proximal part of the leading edge. When the lengths of the 4 segments were added, a wing length of 0.364 m resulted. This value was used for further analysis. We worked in a similar way in the other cases.

Wingspan b (unit m): the sum of the length of the left- and right-wing lengths plus body width.

Wing chord c (unit m): the distance between the leading and the trailing (posterior) edge of the wing. The wing chord was measured perpendicular to the long axis of the wing from the leading edge to the trailing edge of the wing. Wing chord changes from the base to the tip of a wing. It was typically determined at about 40% of wing length as calculated from the base of the wing (see dashed black line in Figure 1b). 

Aspect ratio AR: AR = b/c. 

Wing area S (unit m^2^): The wing length and wing chord were first multiplied for each wing separately to arrive at S_r_ = l∙c. Here, the subscript “r” refers to the fact that S_r_ represents a *rectangular* area. Since the wing chord is, however, not constant along the long axis of the wing, S_r_ has a larger value than the true wing area S. A correction factor was calculated from a wing that was stretched as in flapping flight (Figure 1a). From the photo shown in Figure 1a, we also derived the orientation of the primaries during flight. This is shown by the marking of the primaries in Figure 1a. We then aligned the leading edge of the wing as it occurred in the flying bird with the leading edge of a photograph of a stretched wing of a hand-held bird (Figure 1b). The alignment demonstrated that the outer primaries in the hand-held bird were not correctly stretched (Figure 1b). We found a correction factor of 12% for wing area for both the wing from the flying bird and the stretched wing after correcting for the orientation of the primaries. Thus, wing area S was estimated to be S_r_∙(1−0.12). We applied this correction to every wing examined.

Body mass bm (unit kg): This parameter was measured to the closest 0.005 kg. We are aware that the body mass of owls may undergo considerable changes. In the birds caught in the field, the actual measured body mass was used. If body mass was taken more than once in a bird, the results were averaged (see Appendix A). In the captive birds, the mean free-feeding body mass as it occurred in the course of time (often more than one year) was used.

Wing loading WL (unit N/m^2^, with N = Newton): WL = bm/S. Here we deviate from using SI and use force, measured in Newton (N), as is performed in most other studies.

For the judgement of the scaling of the aerodynamic parameters, the predictions for the parameters in the case of isometry are available (see, e.g., [24]). In the simplest model, the parameters with the dimension of a length (x) should scale with body mass as x~bm^0.33^. This should also hold for wing loading, while the aspect ratio is dimensionless and should be independent of body mass. The parameters with the dimension of an area (y) should scale with body mass as y~bm^0.67^. We plotted the relations in a log–log plot and derived the slopes that correspond, in such plots, to the exponent of the non-logarithmic equations. The data from the whole samples, but also separately for males and females, were obtained.

We used non-parametric statistics as a measure of caution because, with the low number of cases, it is difficult to prove that data are normally distributed. Non-parametric statistics do not have such a requirement. The statistical significance between the *T.f.pratincola* and *T.f.punctatissima* samples and between the sexes in each of these samples was checked with a Mann–Whitney U-test. Such a test was not carried out for wingspan because, due to the definition, the statistics are the same as with wing length. A Wilcoxon Signed Rank test was used to test the independence of the measurements on the left and right wings. Confidence intervals for the slopes in the log–log plots were calculated to obtain insight into whether the measured slopes were different from the predictions outlined in the last paragraph or not. 

## 3. Results

Since we typically took measurements from both wings of an animal, the question arose whether the left and right wings may be treated as independent measures. A comparison of the lengths, chords, and areas of the left and right wings in the seven *T.f.punctatissima* specimens for which we have data for both wings did not allow us to reject the null hypothesis (Wilcoxon Signed Rank test, each *p*-value > 0.5, each z-score < 0.6). Therefore, we averaged the data from the two wings and included only the averaged value in the following analyses.

### 3.1. Differences between the Two Subspecies

The subspecies *T.f.pratincola* and *T.f.punctatissima* are distinctly different in size, with *T.f.pratincola* having a body mass approximately twice that of *T.f.punctatissima* (Table 1). The body-mass difference is highly significant (Table 1). The size difference is also reflected in the wing parameters. All of the tested wing parameters, apart from the aspect ratio, are highly significantly different (Table 1). For example, if expressed in numbers, in *T.f.pratincola*, the wing is almost 20% longer than in *T.f.punctatissima* (Table 1), as is the wingspan (Table 1). By contrast, the wing chord in *T.f.punctatissima* is only about 10% shorter than in *T.f.pratincola* (Table 1). Nevertheless, the difference is highly significant. The size differences in wing length and wing chord result in a broader wing in *T.f.punctatissima* compared with *T.f.pratincola* (Figure 2). This is reflected by a larger factor wing chord/wing length in *T.f.punctatissima* compared with *T.f.pratincola* (Table 1), and a wing area in *T.f.punctatissima* that is only 74% of that in *T.f.pratincola* (Table 1). The only parameter that is just significantly, but not highly significantly different, is the aspect ratio (Table 1), with *T.f.pratincola* having a higher aspect ratio than *T.f.punctatissima*.

Because the two species are distinctly different in size, as demonstrated in Table 1, we wondered how do important aerodynamic parameters scale. In these tests, we used predictions for scaling mentioned in the introduction. To obtain information about variability, we included the 95% and 99% confidence intervals of the scaling factors (Table 2).

The scaling of wing length, as measured by the slope in the log–log plot, is not different from isometry) (Table 2, expected: 0.33, measured: 0.29, 95% confidence interval: 0.222–0.358). By contrast, the slope for the wing chord was much lower than expected. The 99% confidence interval of the measured slope did not include the predicted slope (Table 2). The relatively longer wing chord in *T.f.punctatissima* is also the main reason why the wing area scales with a lower value than expected. This deviation from the expected value is also highly significant (Table 2). The relatively larger wing area leads to a much lower wing loading in *T.f.punctatissima* than in *T.f.pratincola* (Figure 3, Table 1). The mean value for *T.f.punctatissima* is lower than expected from isometry, as demonstrated by the higher slopes than expected (Table 2, Figure 3). The 99% confidence interval for the measured slope does not include the predicted value. Thus, wing loading is also highly significantly different from isometry. In fact, the wing-loading values of the two populations do not overlap (Figure 3). In other words, the highest value of wing loading in the *T.f.punctatissima* sample is lower than the lowest value of wing loading in the *T.f.pratincola* sample. For an aspect ratio that is about 7% lower in *T.f.punctatissima* than in *T.f.pratincola* (Table 1) with a significant difference, the scaling factor is positive and different from the expected value (0) for the 95% criterion, but not for the 99% criterion (Table 2). 

Overall, the comparison of the data from *T.f.pratincola* and *T.f.punctatissima*, as shown in Table 1 and Table 2 as well as in Figure 2 and Figure 3, demonstrate allometric scaling for wing chord, wing area, wing loading, and aspect ratio. By contrast, the scaling of wing length is not statistically different from the value expected for isometry.

### 3.2. Differencees between the Sexes

Raptors, including owls, are known to exhibit sex differences. Therefore, in the next step, we checked for differences between males and females (Figure 3, Table 3 and Table 4). The only significant difference we observed pertained to body mass in *T.f.pratincola* (Table 3). Males of *T.f.pratincola* were lighter by about 16% than females. The body mass difference in *T.f.punctatissima* was 10%, with a trend that females are heavier than males. However, the difference was not significant (Table 4). Since we have data from more *T.f.punctatissima* owls, which did not qualify to be included in this study because of missing wing data, we also tested whether there was a significant difference if we included more birds (see Appendix A). This was indeed the case. The trend seen in Table 4 for body mass was changed to a statistically significant difference if we included data from all of the birds that we examined (four males, seven females; Mann–Whitney U test, U = 1.5; z = 2.2782; *p* = 0.02272). The few field measurements (three birds) in successive years and different times of the year in *T.f.punctatissima* yielded body-mass differences of up to 15% but not more (Appendix A). This suggested to us that body mass can well be captured by one-time measurements in the field. In both subspecies, the wing parameters examined were not statistically different, and there was also no trend (*p* < 0.1) seen in the analyses (Table 3 and Table 4). This is also obvious in Figure 3, in which the wing loading is plotted separately for males and females as a function of body mass. 

Thus, there was a sex difference with respect to body mass, but not for the wing parameters examined.

## 4. Discussion

### 4.1. Methodological Considerations

We were aware that measuring wing parameters is a difficult task (see, e.g., [25]). This holds especially for determining wing length. Wing length is influenced crucially by wing stretching. We used the wing shape of a flying bird as a reference for determining wing length, wing chord, and wing area from our images because we argue that in flight, the wing takes the most natural shape. The congruence of our data with those of others (see next section) also suggests that our measurement methods veridically represent the differences in the parameters analyzed.

In this study, we did not consider the influence of the tail and the tail feathers on barn-owl flight because we did not have data on feather spreading and tail movement of these birds during flight. The inspection of photographs shows that, especially during landing, tail feathers are widely spread and influence aerodynamic performance [4]. Therefore, it would be interesting to quantify the contribution of the tail to barn-owl flight in the future. We also did not take into account the molting of flight feathers. Since barn owls molt sequentially, we argue that molting, although it might increase the variability of the data, would not change the main conclusions.

### 4.2. Comparison with Other Studies

Reversed sexual size dimorphism, the observation that females of a species are larger than males, is common in owls and raptors [26]. We observed this phenomenon as well with respect to body mass in both subspecies. With respect to scaling, there is both isometric and allometric scaling of flight-related parameters (e.g., [14,27,28]). For owls, data on 15 owl species from North America were reported in [14], while [29] summarized data on nine European species (see Appendix A for specifics). When we compared our data on wing loading to the data published in these two reports, at first view, nothing appeared conspicuous (Figure 4). However, a closer look reveals that the slope relating wing loading with body mass in the different American or European owl species is much lower (0.159 in [14], 0.231 in [29]) than the slope in the barn owls we studied (0.559) (Figure 4). The European barn owl (*Tyto alba*), which is intermediate in size compared with *T.f.pratincola* and *T.f.punctatissima*, had an intermediate wing loading (29 N/m^2^, [29]). The wing loading of the European species fits well into the log–log relation for the two subspecies of *Tyto furcata* that we studied (see arrow in Figure 4). 

Data on the aerodynamic parameters of *T.f.pratincola* are available from [12,13,14], amongst others. The mean body mass and the mean wing areas we found are smaller than those reported by [12] (compare our data with those of [12] in Table 3). However, the male/female relations reported in [12] with respect to body mass, wing area, and wing loading fit our data well (Table 3). Our data also fits well with the data reported in [14]. This could mean that the population from California studied here and in [14] is lighter in body mass than the population from Northern Utah studied in [12]. In addition, our data on the wing area of male *T.f.pratincola* are close to the wing area reported in [14] for one *T.f.pratincola* male. With respect to wing loading, the data we report here for *T.f.pratincola* also fits the wing-loading data reported earlier from our laboratory well [13]. Our data are, however, about 10% higher than those reported by Marti [12]. The similarity of the data reported here with data reported earlier suggests that our sample, albeit small, well captures the situation in the population of *T.f.pratincola* if one keeps in mind that some variation may exist in the specimen occurring across the large North American continent. We like to think that the data of *T.f.punctatissima* we present here also captures well the situation in this subspecies. This conclusion is supported by data in [21]. For example, the mean body mass we determined in female *T.f.punctatissima* (0.257 kg for four females, 0.264 kg for seven females) is very close to the body mass de Groot [21] reported (0.264 kg). Moreover, this author also reported less sexual dimorphism in *T.f.punctatissima* than in other barn-owl species. Despite these consistencies, it would be interesting to study more specimens of both subspecies to find out whether aerodynamic parameters differ as in [12] or not.

### 4.3. Speculations on the Evolutionary Basis of the Lower Wing Loading in T.f.punctatissima

The subspecies *T.f.punctatissima* stood out in that it had (1) a lower wing loading than expected from the relation seen in other owl species (the other data point in Figure 4 showing a very low wing loading (21 N/m^2^) is from the long-eared owl, with a comment by the author that this may be an artifact (p. 555 in [14])), and (2) a very small reversed sexual dimorphism with respect to wing loading. We speculate that the missing difference with respect to wing loading in *T.f.punctatissima* is due to evolutionary pressure not only for males but also for females to keep wing loading low. Smaller values of wing loading correspond to higher maneuverability in *T.f.punctatissima* compared with *T.f.pratincola*. The population of *T.f.pratincola* studied in [22] consumed nearly 100% of the vertebrate prey. Although the diet of *T.f.punctatissima* consists of about 85% small rodents, in many pellets, insects are found as well [21,30]. This might mean that *T.f.punctatissima*, in contrast to *T.f.pratincola*, includes smaller prey in its diet, and capturing small prey may be easier with lower wing loading. This hypothesis should be, however, taken with caution because most of the insect prey of *T.f.punctatissima* were Tettigoniidae [21] that sing during the night but are stationary while doing so and that—at least today—small rodents are abundant on Galapagos. However, house mice in Galapagos clearly have a lower body mass than those in central Europe (Wagner, pers. observation). No information is available on how the situation was before humans introduced mice, rats, and invertebrates to the Galapagos, and the owls had to prey on rice rats (that are now extinct on most islands) and endemic insects. It would be interesting to examine whether food availability played a role in developing low-wing loading in *T.f.punctatissima*. Likewise, it would be interesting whether the indication of miniaturization in *T.f.punctatissima* contributed to the low-wing loading by comparing the situation in the barn owl with other closely related species in which one part lives on the remote islands.

## 5. Conclusions

What may this result mean? On the one hand, there may be a special scaling for the three barn-owl taxa we examined. On the other hand, *T.f.punctatissima* may be an outlier. We cannot discriminate between these two hypotheses. However, both results would be interesting. The earlier would point to a special adaptation in the genus Tyto, and the latter would argue for a special adaptation in a subspecies living on a remote island. To get better insight in such issues, it would be interesting to study more barn-owl species and more specimens of the two subspecies examined here.

## Figures and Tables

**Figure 1 animals-12-02532-f001:**
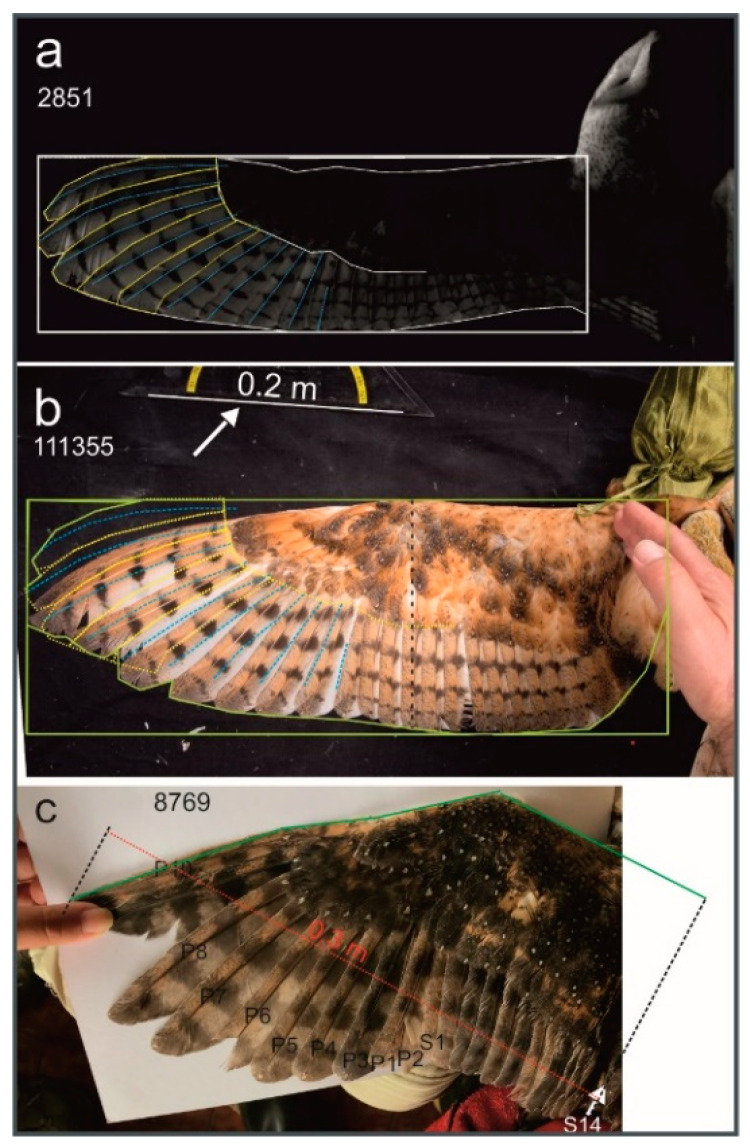
Methods to reconstruct wing parameters. (**a**) Image extracted of a video that shows wing spreading as it occurs in flight (*T.f.pratincola*). The distal primaries and the rachises are marked in yellow and blue, respectively. (**b**) Wing of a living *T.f.pratincola* that was stretched while held by hand. Note that the distal primaries were not appropriately spaced, as may be seen by comparing the marked feather outlines taken from (**a**). The arrow points to the ruler that was used for scaling (white line corresponds to 0.20 m). The dashed black line represents chord length. (**c**) Photograph of a wing of a *T.f.punctatissima*. The wing was measured with a ruler (indicated by the red line) to have a length of 0.3 m. The different segments (green lines) and their addition demonstrated that veriical wing length was 0.364 m. The black dashed line on the right outlines the orientation of the body and the base of the wing. Px: primaries, Sx: secondaries; 2851, 111355, 8679: identification codes for the photos.

**Figure 2 animals-12-02532-f002:**
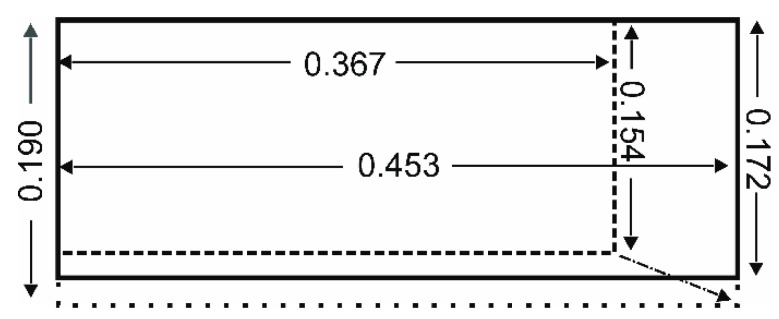
Comparison of wing shape in the two subspecies. The outline of the wing of *T.f.pratincola* is drawn by solid lines, that of *T.f.punctatissima* by dashed lines (original size) and by dotted lines (stretched to the length of *T.f.pratincola*). The numbers represent mean values in meters (see Table 1). Note the relatively longer chord in *T.f.punctatissima* compared with *T.f.pratincola*.

**Figure 3 animals-12-02532-f003:**
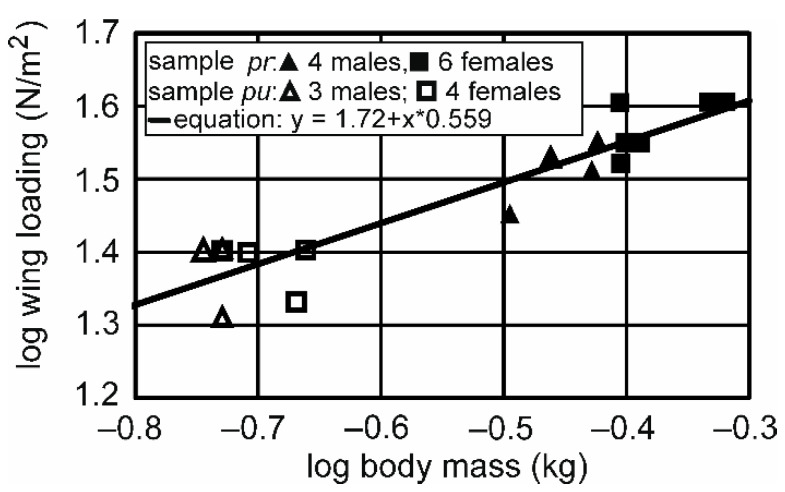
Scaling of wing loading with body mass. The solid black line indicates the linear relation in the log–log plot with the slope (0.559) corresponding to the exponent of the non-logarithmized equation. Note that the scaling is clearly allometric (see also Table 2).

**Figure 4 animals-12-02532-f004:**
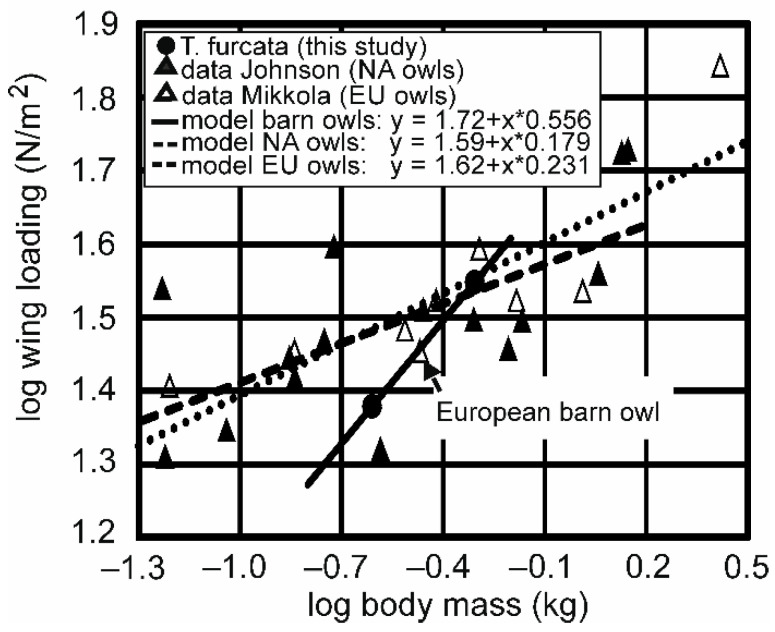
Comparison of wing-loading data. The data of the other owl species were taken from [14,29] (for numbers, see Appendix A). Note that the relation between wing loading and body mass exhibits a much lower slope if all owl species are considered than when only the barn owls are taken into account.

**Table 1 animals-12-02532-t001:** Morphometric characteristics of Barn-owl wings: Data of both subspecies.

Characteristic	Tyto furcata ^1^ pratincola (pr)	Tyto furcata ^1^ punctatissima (pu)	Factor pu/pr	Statistics ^2^
Number of animals	10	7		
Wing length (m)	0.453 ± 0.01	0.367 ± 0.02	0.81	U = 0; z = 3.3689; *p* = 0.0007456
Wingspan (m) ^3^	1.01 ± 0.01	0.813 ± 0.05	0.81	Follows wing length
Chord length (m)	0.172 ± 0.01	0.154 ± 0.01	0.90	U = 6.5; z = 2.7359; *p* = 0.006221
Total wing area (m^2^)	0.1378 ± 0.009	0.1015 ± 0.013	0.74	U = 1; z = 3.2713; *p* = 0.001071
Body mass (kg)	0.498 ± 0.061	0.246 ± 0.019	0.49	U = 0; z = 3.751; *p* = 0.000738
Wing loading (N/m^2^)	35.5 ± 3.9	24 ± 2.1	0.68	U = 0; z = 3.731; *p* = 0.000743
Chord/length c/l	0.381 ± 0.01	0.417 ± 0.03	1.09	U = 4; z = 2.9783; *p* = 0.002898
Aspect ratio	5.73 ± 0.1	5.35 ± 0.4	0.93	U = 12; z = 2.1985; *p* = 0.0297

^1^ Data are presented as means and standard deviation; ^2^ Mann–Withney U-Test: https://www.statskingdom.com/170median_mann_whitney.html (accessed from March to September 2022); ^3^ with body width 1 m for *T.f.pratincola* and 0.08 m for *T.f.punctatissima.*

**Table 2 animals-12-02532-t002:** Scaling of parameters measured in *T.f.pratincola* and *T.f.punctatissima*.

Parameter	Slope ^1^	Slope	CI ^2^ of Slope		CI of Slope	
	Predicted	Measured	95% CI		99% CI	
			lower	upper	lower	upper
Wing length	0.333	0.290	0.222	0.358	0.196	0.384
Chord length	0.333	0.162	0.088	0.235	0.060	0.264
Total area	0.667	0.441	0.319	0.564	0.272	0.611
Wing loading	0.333	0.559	0.436	0.681	0.389	0.728
Chord/length	0	−0.116	−0.186	−0.045	−0.213	−0.018
Aspect ratio	0	0.090	0.022	0.158	−0.004	0.184

^1^ Slope in model: parameter = f (body mass)^slope^, ^2^ CI = confidence interval.

**Table 3 animals-12-02532-t003:** Morphometric characteristics of Barn-owl wings: Data of *Tyto furcata pratincola*, including data in [12].

Characteristic ^1^	Male (m)	Female (f)	Factor m/f	Statistics
Number of animals	4	6		
Wing length (m)	0.451 ± 0.01	0.454 ± 0.02	0.99	U = 9; z = 0.5108; *p* = 0.6095
Wingspan (m)	1.00 ± 0.02	1.01 ± 0.03	0.99	follows wing length
Chord length (m)	0.169 ± 0.01	0.174 ± 0.01	0.97	U = 5; z = 1.39: *p* = 0.1645
Total wing area (m^2^) Data in [12] ^2^	0.1348 ± 0.008 0.1576	0.1397 ± 0.010 0.1663	0.96 0.95	U = 6; z = 1.1762; *p* = 0.2395
Body mass (kg) Data in [12] ^2^	0.446 ± 0.034 0.474	0.533 ± 0.049 0.566	0.84 0.84	U = 0; z = 2.5927; *p* = 0.00952
Wing loading (N/m^2^) Data in [12] ^2^	32.5 ± 3.1 29.3	37.5 ± 3.2 33.4	0.87 0.88	U = 5; z = 1.3943; *p* = 0.1632
Chord/length	0.376 ± 0.01	0.383 ± 0.01	0.98	U = 6; z = 1.1762; *p* = 0.2395
Aspect ratio	5.80 ± 0.1	5.68 ± 0.1	1.02	U = 5.5; z = 1.287; *p* = 0.1981

^1^ for details see legend to Table 1, ^2^ data from [12] are presented as mean values as appearing in the publication.

**Table 4 animals-12-02532-t004:** Morphometric characteristics of Barn-owl wings: Data of *Tyto furcata puncatissima*.

Characteristic ^1^	Male (m)	Female (f)	Factor m/f	Statistics
Number of animals	3	4		
Wing length (m)	0.364 ± 0.02	0.369 ± 0.03	0.99	U = 5.5; z = 0; *p* = 1
Wingspan (m)	0.81 ± 0.05	0.82 ± 0.05	0.99	follows wing length
Chord length (m)	0.151 ± 0.01	0.157 ± 0.01	0.96	U = 4; z = 0.4837; *p* = 0.6286
Total wing area (m^2^)	0.0973 ± 0.014	0.1046 ± 0.014	0.93	U = 3; z = 0.8416; *p* = 0.4
Body mass (kg)	0.232 ± 0.005	0.257 ± 0.019	0.90	U = 1; z = 1.651; *p* = 0.09873
Wing loading (N/m^2^)	23.7 ± 2.8	24.2 ± 1.9	0.98	U = 5.5; z = 0; *p* = 1
Chord/length c/l	0.413 ± 0.02	0.419 ± 0.04	0.99	U = 5; z = 0.18; *p* = 0.8571
Aspect ratio	5.39 ± 0.3	5.32 ± 0.4	1.01	U = 5; z = 0.18; *p* = 0.8571

^1^ for details see legend to Table 1.

## Data Availability

Data are available from the corresponding author.

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
