# Peer review of "A Comparison of Aerodynamic Parameters in Two Subspecies of the American Barn Owl (Tyto furcata)"

_animals, 2022, doi:10.3390/ani12192532_

Round 1
Reviewer 1 Report
Review of Wagner and Piedrahita submitted to “animals”
I have no major concerns of substance.
According to the internet, “Tyto furcata” is often considered a subspecies of Tyto alba
How was the subspecies punctatissima determined? Did the authors do a formal analysis of characaters, or did they simply assume the Galapagos birds were puntiatissima because the birds were captured in the galapagos? (that assumption seems fine to me but I suggest stating it in the methods)
Line 51: is this a case of island minutism? Do other island populations of barn owl show the same thing?
Line 65-70: captive individuals may not be the best reference material. Wild animals would be better to include, since captive breeding is well known to sometimes cause captive animals to evolve differences relative to their progenitors (Such as higher body weight)
The wing in figure 1C –while the issue of the incomplete spread of the wing is addressed below, was this bird molting? that was my first impression of the gap in the outer primaries as shown. I’m concerned molt could affect estimates of wing loading, which is the main result of this study (a word search suggests molt or moult are not mentioned in the text)
Line 97: clarify what the “base” is. Do you mean the shoulder joint?
Line 134: I can’t see a green dashed line in Fig. 1B (make it bigger?)
Reviewer 2 Report
I carefully read the manuscript entitled “A comparison of aerodynamic parameters in two subspecies of the American barn owl (Tyto furcata) in relation to other owl and bird species”.
Overall, the manuscript has the potential to provide insights into the aerodynamic parameters of the two subspecies. However, the data come from very few individuals and the comparison comes from a captive and a wild population using photographs and direct measurements of the wing parameters. The ms should be more informative. Below my general and some specific comments on critical points.
General comments
The title should be different eg. delete “in relation…species”. There is no comparison to other owl and bird species except from a paragraph in the discussion. See my comments below.
Introduction: authors should make a more detailed literature review and give more details on the way the morphological specializations on the feathers and wing as well as the low-speed flight affect the silent flight.
Please give a separate paragraph about the importance of barn owl in studying flight and give details about the species Tyto furcata including the two subspecies and their differences. In lines 44-47 should give some details on the way these parameters affect the silent flight of the European barn owl. What abouts other owl species and silent flight?
The authors should follow a standard reference on the two subspecies. In the text they refer as: Tyto furcata punctatissima, as only punctatissima, as Galapagos barn owl or as sample “G”. please use only one reference along the text in order to be more easily readable. I would suggest to refer to either the North American and the Galapagos barn owl.
Please add a paragraph that states clearly the aim of this study.
Materials and Methods: the authors should not use “more than once” and should mention exactly how many times they took data on wings and body mass from each bird during these years (lines75-76). Also, did the authors examined if these measurements differed?
Rewrite the following sentence in lines 82-85 as: “We were interested in the following aerodynamic parameters of owl flight:”. Each parameter is analyzed below so you can state there the abbreviation and the units and not repeat them. For example in line 97 you should write “Wing length (l) in meters (m):……”.
Please explain the statistical analyses were used to test differences between the sexes, between the subspecies or between all the above? Also, explain why you used non parametric statistics?
Results: this section must be clearer to the reader. Please rewrite the Table 1 and include all the statistics derived from the comparisons. Also, please rewrite all the values in both tables by adding the 0 before the full stop. The caption of the tables should be more informative and give all the details as they can stand alone in the ms. In lines 173-178 please define which wing did you finally used in the analyses and present the statistics. If they are insignificant, please make a table in the appendix.
Line 180: as I mentioned above, please do not use just the subspecies (eg. pratincola) and use a standard reference.
Lines 237-238: in my opinion the result section should not include references and information that should be in the discussion section.
Discussion: lines 264-265: the spreading of the flight feather derived from a flying bird only for the captive individuals. Please define it.
Line 275: please define about the RSD you observed. In what measurements?
The subsection “comparison with other studies” should followed by a subsection in the methodology and the results and by adding a table that presents all the values derived for every species from the literature and the results taken from this study. For me it is difficult to follow the information given here.
Specific comments
Line 40: substitute “spec” with “sp”.
Line 128: delete the second “Wingspan b”
Line 130: delete the second “Wing chord c”
Line 149: delete the second “Body mass”
Reviewer 3 Report
All parts of this manuscript were described in details. The aim of the study is clear and the obtained results were supported by the correct figures. Thus in my opinion this manuscript can be accept in present form.
Author Response
We appreciate this response and thank the reviewer for his words.
Round 2
Reviewer 2 Report
I am pleased that the authors addressed most of my comments and the ms has been improved. However, I have some last comments.
The introduction section should be more detailed. In the first paragraph the authors refer to the aerodynamic adaptations of barn owls that make these species interesting in studying but they do not explain why. “the European barn owl (Tyto alba) is in the lower range compared with other bird species”…..”Moreover, body mass of barn owls is comparatively low. By contrast, wing area in these birds is relatively large, while aspect ratio is low.”……why these are important? Please give some more details.
The authors have changed the abbreviation of both subspecies in pr and pu accordingly. In my opinion this more confusing. As the authors want to use the scientific name of the subspecies then to refer to them as T.f.pratincola and T.f.punctatissima in all section of the ms and not as in the first draft that used only the naming of the subspecies.
In pages 10-11 there are 3 copies of figure 4. Please delete the 2 of them.
In line 437 I believe the scaling involved thee barn owl subspecies (European, North American, Galapagos) and not species.
